# Deep Model Reassembly

**Xingyi Yang**[1]    **Daquan Zhou**[1,2]    **Songhua Liu**[1]    **Jingwen Ye**[1]    **Xinchao Wang**[1]
[1]National University of Singapore    [2]Bytedance
{xyang,daquan.zhou,songhua.liu}@u.nus.edu, {jingweny,xinchao}@nus.edu.sg

## Abstract

In this paper, we explore a novel knowledge-transfer task, termed as Deep Model Reassembly (DeRy), for general-purpose model reuse. Given a collection of heterogeneous models pre-trained from distinct sources and with diverse architectures, the goal of DeRy, as its name implies, is to first dissect each model into distinctive building blocks, and then selectively reassemble the derived blocks to produce customized networks under both the hardware resource and performance constraints. Such ambitious nature of DeRy inevitably imposes significant challenges, including, in the first place, the feasibility of its solution. We strive to showcase that, through a dedicated paradigm proposed in this paper, DeRy can be made not only possibly but practically efficient. Specifically, we conduct the partitions of all pre-trained networks jointly via a cover set optimization, and derive a number of *equivalence set*, within each of which the network blocks are treated as functionally equivalent and hence interchangeable. The equivalence sets learned in this way, in turn, enable picking and assembling blocks to customize networks subject to certain constraints, which is achieved via solving an integer program backed up with a training-free proxy to estimate the task performance. The reassembled models, give rise to gratifying performances with the user-specified constraints satisfied. We demonstrate that on ImageNet, the best reassemble model achieves $78.6\%$ top-1 accuracy without fine-tuning, which could be further elevated to $83.2\%$ with end-to-end training. Our code is available at https://github.com/Adamdad/DeRy.

## 1 Introduction

The unprecedented advances of deep learning and its pervasive impact across various domains are partially attributed to, among many other factors, the numerous *pre-trained models* released online. Thanks to the generosity of our community, models of diverse architectures specializing in the same or distinct tasks can be readily downloaded and executed in a plug-and-play manner, which, in turn, largely alleviates the model reproducing effort. The sheer number of pre-trained models also enables extensive knowledge transfer tasks, such as knowledge distillation, in which the pre-trained models can be reused to produce lightweight or multi-task students.

In this paper, we explore a novel knowledge transfer task, which we coin as *Deep Model Reassembly* (DeRy). Unlike most prior tasks that largely focus on reusing pre-trained models as a whole, DeRy, as the name implies, goes deeper into the building blocks of pre-trained networks. Specifically, given a collection of such pre-trained heterogeneous models or *Model Zoo*, DeRy attempts to first dissect the pre-trained models into building blocks and then reassemble the building blocks to tailor models subject to users' specifications, like the computational constraints of the derived network. As such, apart from the flexibility for model customization, DeRy is expected to aggregate knowledge from heterogeneous models without increasing computation cost, thereby preserving or even enhancing the downstream performances.

36th Conference on Neural Information Processing Systems (NeurIPS 2022).

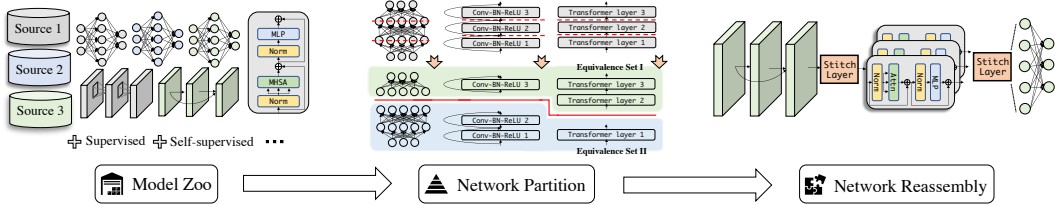

**Figure 1:** Overall workflow of `DeRy`. It partitions pre-trained models into equivalent sets of neural blocks and then reassemble them for downstream transfer. Both steps are optimized through solving constrained programs.

Admittedly, the nature of `DeRy` *per se* makes it a highly challenging and ambitious task; in fact, it is even unclear whether a solution is feasible, given that no constraints are imposed over the model architectures in the model zoo. Besides, the reassembly process, which assumes the building blocks can be extracted in the first place, calls for a lightweight strategy to approximate the model performances without re-training, since the reassembled model, apart from the parametric constraints, is expected to behave reasonably well.

We demonstrate in this paper that, through a dedicated optimization paradigm, `DeRy` can be made not only possible by highly efficient. At the heart of our approach is a two-stage strategy that first partitions pre-trained networks into building blocks to form *equivalence sets*, and then selectively assemble building blocks to customize tailored models. Each equivalence set, specifically, comprises various building blocks extracted from heterogeneous pre-trained models, which are treated to be functionally equivalent and hence interchangeable. Moreover, the optimization of the two steps is purposely decoupled, so that once the equivalence sets are obtained and fixed, they can readily serve as the basis for future network customization.

We show the overall workflow of the proposed `DeRy` in Figure 1. It starts by dissecting pre-trained models into disjoint sets of neural blocks through solving a cover set optimization problem, and derives a number of equivalence sets, within each of which the neural blocks are treated as functionally swappable. In the second step, `DeRy` searches for the optimal block-wise reassembly in a training-free manner. Specifically, the transfer-ability of a candidate reassembly is estimated by counting the number of linear regions in feature representations [55], which reduces the searching cost by $10^4$ times as compared to training all models exhaustively.

The reassembled networks, apart from satisfying the user-specified hard constraints, give rise to truly encouraging results. We demonstrate through experiments that, the reassembled model achieves $>$ 78% top-1 accuracy on Imagenet with all blocks frozen. If we allow for finetuning, the performances can be further elevated, sometimes even surpassing any pre-trained network in the model zoo. This phenomenon showcases that `DeRy` is indeed able to aggregate knowledge from various models and enhance the results. Besides, `DeRy` imposes no constraints on the network architectures in the model zoo, and may therefore readily handle various backbones such as CNN, transformers, and MLP.

Our contributions are thus summarized as follows.

1. We explore a new knowledge transfer task termed Deep Model Reassembly (`DeRy`), which enables reassembling customized networks from a zoo of pre-trained models under user-specified constraints.

2. We introduce a novel two-stage strategy towards solving `DeRy`, by first partitioning the networks into equivalence sets and then reassembling neural blocks to customize networks. The two steps are modeled and solved using constrained programming, backed up with training-free performance approximations that significantly speed up the knowledge-transfer process.

3. The proposed approach achieves competitive performance on a series of transfer learning benckmarks, sometimes even surpassing than any candidate in the model zoo, which, in turn, sheds light on the the universal connectivity among pre-trained neural networks.

## 2 Related Work

**Transfer learning from Model Zoo.** A standard deep transfer learning paradigm is to leverage a single trained neural network and fine-tune the model on the target task [85, 87, 45, 35, 98, 96, 33]

| Problem | No need to retrain | Adaptive Architecture | No Additional Computation | Utilize All Models | Heterogeneous Architecture |
|---|---|---|---|---|---|
| Single Model Transfer | ✓ | ✗ | ✓ | ✗ | ✗ |
| Zoo Transfer by Selection | ✓ | ✗ | ✓ | ✗ | ✓ |
| Zoo Transfer by Ensemble | ✓ | ✗ | ✗ | ✓ | ✓ |
| Zoo Transfer by Parameter Fusion | ✓ | ✗ | ✓ | ✓ | ✗ |
| Neural Architecture Search | ✗ | ✓ | - | - | - |
| DeRy | ✓ | ✓ | ✓ | ✓ | ✓ |

**Table 1:** Comparison of a series of transfer learning tasks and our proposed Deep Model Reassembly.

or impart the knowledge to other models [31, 88, 70, 90, 91, 89, 48]. The availability of large-scale model repositories brings about a new problem of transfer learning from a model zoo rather than with a single model. Currently, there are three major solutions. One line of works focuses on select one best model for deployment, either by exhaustive fine-tuning [39, 74, 87] or quantifying the model transferability [94, 92, 57, 76, 4, 73, 76, 6, 41] on the target task. However, due to the unreliable measurement of transferability, the best model selection may be inaccurate, possibly resulting in a suboptimal solution. The second idea was to apply ensemble methods [19, 99, 2, 97], which inevitably leads to prohibitive computational costs at test time. The third approach is to adaptively fuse multiple pre-trained models into a single target model. However, those methods can only combine identical [71, 18, 78] or homogeneous [72, 58] network structures, whereas most model zoo contains diverse architectures. In contrast to standard approaches in Table 1, DeRy dissects the pre-trained models into building blocks and rearranges them in order to reassemble new pre-trained models.

**Neural Representation Similarity.**    Measuring similarities between deep neural network representations provide a practical tool to investigate the forward dynamics of deep models. Let $\mathbf{X} \in \mathbb{R}^{n \times d_1}$ and $\mathbf{Y} \in \mathbb{R}^{n \times d_2}$ denote two activation matrices for the same $n$ examples. A neural *similarity index* $s(X, Y)$ is a scalar to measure the representations similarity between $X$ and $Y$, although they do not necessarily satisfy the triangle inequality required of a proper metric. Several methods including linear regression [86, 31], canonical correlation analysis (CCA) [65, 27, 64], centered kernel alignment (CKA) [40], generalized shape metrics [81]. In this study, we leverage the representations similarity towards function level to quantify the distance between two neural blocks.

**Neural Architecture Search.** Automatic neural archtecture search (NAS) has achieved promising performance-efficiency trade-offs as well as reducing human efforts. With a pre-defined search space [75, 47, 82, 69], the designing problem of the optimal architecture is formalized as a discrete optimization, where the best solution could be found with reinforcement learning (RL) [100], evolutionary algorithms (EA) [66] or gradient-based search [47]. Because it is costly to measure the performance of each candidate, several surrogate methods like one-shot NAS [62, 5, 47], predictor-based NAS [46, 51, 80] and zero-shot NAS [55, 9, 1] are proposed to accelerate the evaluation process. In this paper, we similarly formalize the network reassembly as a search problem; However, compared with NAS that searches at random initialization, DeRy is searching the optimal structure combination along side with network weights. In addition, the search space of DeRy is not preset heuristically, but determined by the network partition results.

**Network Stitching.** Initially proposed by [44], model stitching aims to "plug-in" the bottom layers of one network into the top layers of another network, thus forming a stitched network [3, 16]. It provides an alliterative approach to investigate the representation similarity and invariance of neural networks. A recent line of work achieves competitive performance by stitching a visual transformer on top of the ResNet [74]. Instead of stitching two identical-structured networks in a bottom-top manner, in our study, we investigate to assemble arbitrary pre-trained networks by model stitching.

# 3   Deep Model Reassembly

In this section, we dive into the proposed DeRy. We first formulate DeRy, and then define the functional similarity and equivalent sets of neural blocks to partition networks by maximizing overall groupbility. The resulting neural blocks are then linked by solving an integer program.

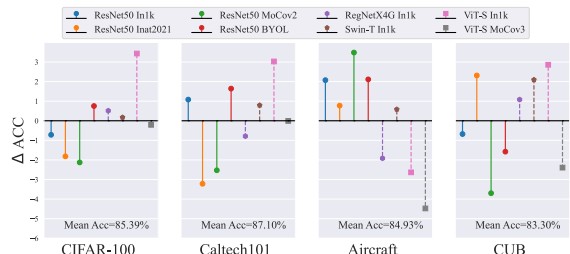

**Figure 2:** The top-1 accuracy difference between "off-the-shelf" pre-trained models on 4 down-stream tasks.

**Table 3:** Accuracy on CIFAR-100 with the pre-trained networks and their reassembled ones.

| Backbone | Init. | #Params(M) | Acc(%) |
|---|---|---|---|
| ResNet50 | in1k sup | 23.71 | 84.67 |
| | inat2021 sup | 23.71 | 82.57 |
| ResNet50 | inat2021(Stage 1&2) in1k(Stage 3&4) | 23.98 | **85.30** |
| ResNet50 Swin-T | in1k sup | 23.71 27.60 | 84.67 85.56 |
| ResNet50(Stage 1&2) Swin-T(Stage 3&4) | in1k sup | 27.94 | **85.77** |

## 3.1 Problem Formulation

Assume we have a collection of $N$ pre-trained deep neural network models $Z = \{\mathcal{M}_i\}_{i=1}^N$ that each composed of $L_i \in \mathbb{N}$ layers of operation $\{F_i^{(k)}\}_{l=1}^{L_i}$, therefore $\mathcal{M}_i = F_i^{(1)} \circ F_i^{(2)} \cdots \circ F_i^{(L_i)}$. Each model can be trained on different tasks or with varied structures. We call $Z$ a *Model Zoo*. We define a learning task $T$ composed of a labeled training set $D_{tr} = \{\mathbf{x}_j, y_j\}_{j=1}^M$ and a test set $D_{ts} = \{\mathbf{x}_j\}_{j=1}^L$.

**Definition 1** *(Deep Model Reassembly) Given a task $T$, our goal is to find the best-performed $L$-layer compositional model $\mathcal{M}^*$ on $T$, subject to hard computational or parametric constraints.*

We therefore formulate it as an optimization problem

$$\mathcal{M}^* = \max_{\mathcal{M}} P_T(\mathcal{M}), \quad s.t. \mathcal{M} = F_{i_1}^{(l_1)} \circ F_{i_2}^{(l_2)} \cdots \circ F_{i_L}^{(l_L)}, |\mathcal{M}| \le C \tag{1}$$

where $F_i^{(l)}$ is the $l$-th layer of the $i$-th model, $P_T(\mathcal{M})$ indicates the performance on $T$, and $|\mathcal{M}| \le C$ denotes the constraints. For two consecutive layers with dimension mismatch, we add a single stitching layer with $1 \times 1$ convolution operation to adjust the feature size. The stitching layer structure is described in Supplementary.

**No Single Wins For All.** Figure 2 provides a preliminary experiment that 8 different pre-trained models are fine-tuned on 4 different image classification tasks. It is clear that no single model universally dominants in transfer evaluations. It builds up our primary motivation to reassemble trained models rather than trust the "best" candidate.

**Reassembly Might Win.** Table 3 compares the test performance between the reassembled model and its predecessors. The bottom two stages of the ResNet50 iNaturalist2021 (inat2021 sup) [77] are stitched with ResNet50 ImageNet-1k (in1k sup) stage 3&4 to form a new model for fine-tuning on CIFAR100. This reassembled model improves its predecessors by 0.63%/2.73% accuracy respectively. Similar phenomenon is observed on the reassembled model between ResNet50 in1k and Swin-T in1k. Despite its simplicity, the experiment provides concrete evidence that the neural network reassembly could possibly lead to better model in knowledge transfer.

**Reducing the Complexity.** From the overall $M = \sum_{i=1}^N L_i$ layers, the search space of Eq 1 is of size L-permutations of M $P(M, L)$, which is undesirably large. To reduce the overall search cost, we intend to partition the networks into blocks rather than the layer-wise-divided setting. Moreover, it is time-consuming to evaluate each model on the target data through full-time fine-tuning. Therefore, we hope to accelerate the model evaluation, even without model training.

Based on the above discussion, the essence of DeRy lies in two steps (1) Partition the networks into blocks and (2) Reassemble the factorized neural blocks. In the following sections, we elaborate on *"what is a good partition?"* and *"what is a good assembly?"*.

## 3.2 Network Partition by Functional Equivalence

A network partition [21, 21] is a division of a neural network into disjoint sub-nets. In this study, we refer specifically to the partition of neural network $\mathcal{M}_i$ along depth into $K$ blocks $\{B_i^{(k)}\}_{k=1}^K$ so that each block is a stack of $p$ layers $B_i^{(k)} = F_i^{(l)} \circ F_i^{(l+1)} \cdots \circ F_i^{(l+p)}$ and $k$ is its stage index. Inspired by the hierarchical property of deep neural networks, we aim to partition the neural networks according to their function level, for example, dividing the network into a "low-level" block that identifies curves and a "high-level" block that recognizes semantics. Although we cannot strictly differentiate "low-level" from "high-level", it is feasible to define *functional equivalence*.

**Definition 2** (*Functional Equivalence*) *Given two functions $B$ and $B'$ with same input space $\mathcal{X}$ and output space $\mathcal{Y}$. $d : \mathcal{Y} \times \mathcal{Y} \to \mathbb{R}$ is the metric defined on $\mathcal{Y}$. For all inputs $\mathbf{x} \in \mathcal{X}$, if the outputs are the equivalent $d(B(\mathbf{x}), B'(\mathbf{x})) = 0$, we say $B$ and $B'$ are functional equivalent.*

A function is then uniquely determined by its peers who generate the same output with the same input. However, we can no longer define functional equivalence among neural networks, since network blocks might have varied input-output dimensions. It is neither possible to feed the same input to intermediate blocks with different input dimensions, nor allow for a mathematically valid definition for metric space [13, 7] when the output dimensions are not identical. We therefore resort to recent measurements on neural representation similarity [27, 40] and define the functional similarity for neural networks. The intuition is simple: two networks are functionally similar when they produces similar outputs with similar inputs.

**Definition 3** (*Functional Similarity for Neural Networks*) *Assume we have a neural similarity index $s(\cdot, \cdot)$ and two neural networks $B : \mathcal{X} \in \mathbb{R}^{n \times d_{in}} \to \mathcal{Y} \in \mathbb{R}^{n \times d_{out}}$ and $B' : \mathcal{X}' \in \mathbb{R}^{n \times d'_{in}} \to \mathcal{Y}' \in \mathbb{R}^{n \times d'_{out}}$. For any two batches of inputs $\mathbf{X} \subseteq \mathcal{X}$ and $\mathbf{X}' \subseteq \mathcal{X}'$ with large similarity $s(\mathbf{X}, \mathbf{X}') > \epsilon$, the functional similarity between $B$ and $B'$ are defined as their output similarity $s(B(\mathbf{X}), B'(\mathbf{X}'))$.*

This definition generalizes well to typical knowledge distillation (KD) [31] when $d_{in} = d'_{in}$, which we will elaborate in the Appendix. We also show in Appendix that Def.3 provides a necessary and insufficient condition for two identical networks. Using the method of Lagrange multipliers, the conditional similarity in Def.3 can be further simplified to $S(B, B') = s(B(\mathbf{X}), B'(\mathbf{X}')) + s(\mathbf{X}, \mathbf{X}')$, which is a summation of its input-output similarity. The full derivation is shown in the Appendix.

**Finding the Equivalence Sets of Neural Blocks.** With Def.3, we are equipped with the math tools to partition the networks into equivalent sets of blocks. Blocks in each set are expected to have high similarity, which are treated to be functionally equivalent and hence interchangeable.

With a graphical notion, we represent each neural network as a path graph $G(V, E)$ [25] with two nodes of vertex degree 1, and the other $n - 2$ nodes of vertex degree 2. The ultimate goal is to find the best partition of each graph into $K$ disjoint sub-graphs along the depth, and the dissected sub-nets are concurrently grouped into $K$ functional equivalence sets, that sub-graph within each group has maximum internal functional similarity $S(B, B')$. In addition, we take a mild assumption that each sub-graph should have approximately similar size $|B_i^{(k)}| < (1 + \epsilon)\frac{|\mathcal{M}_i|}{K}$, where $|\cdot|$ indicates the model size and $\epsilon$ is coefficient controls size limit for each block. We solve the above problem by posing a tri-level constrained optimization with joint clustering and partitioning

$$\max_{B_{a_j}} \quad J(A, \{B_i^{(k)}\}) = \max_{A_{(ik,p)} \in \{0,1\}} \sum_{i=1}^{N} \sum_{j=1}^{K} \sum_{k=1}^{K} A_{(ik,j)} S(B_i^{(k)*}, B_{a_j}) \quad \text{(Clustering)} \quad (2)$$

$$s.t. \quad \sum_{j=1}^{K} A_{(ik,j)} = 1, \quad \{B_i^{(k)*}\}_{k=1}^{K} = \arg\max_{B_i^{(k)}} \sum_{k=1}^{K} A_{(ik,j)} S(B_i^{(k)}, B_{a_j}) \quad \text{(Partition)} \quad (3)$$

$$s.t. \quad B_i^{(1)} \circ B_i^{(2)} \cdots \circ B_i^{(K)} = \mathcal{M}_i, B_i^{(k_1)} \cap B_j^{(k_2)} = \emptyset, \forall k_1 \neq k_2 \quad (4)$$

$$|B_i^{(k)}| < (1 + \epsilon)\frac{|\mathcal{M}_i|}{K}, k = 1, \ldots K \quad (5)$$

Where $A \in \mathbb{N}^{KN \times K}$ is the 0-1 assignment matrix, where $A_{(ik,j)} = 1$ denote the $B_i^{(k)}$ block belongs to the $j$-th equivalence set, otherwise 0. Note that each block only belongs to one equivalence set, thus each column sums up to 1, $\sum_{j=1}^{K} A_{(ik,j)} = 1$. $B_{a_j}$ is the anchor node for the $j$-th equivalence set, which has the maximum summed similarly with all blocks in set $j$. $B_i^{(k_1)} \cap B_j^{(k_2)} = \emptyset$ refers to the fact the no two blocks has overlapping nodes.

The inner optimization largely resembles the conventional *set cover problem* [32] or $(K, 1 + \epsilon)$ *graph partition problem* [36] that directly partition a graph into $k$ sets. Although the graph partition falls exactly in a NP-hard [28] problem, heuristic graph partitioning algorithms like Kernighan-Lin (KL) algorithm [38] and Fiduccia–Mattheyses (FM) algorithm [23] can be applied to solve our problem efficiently. In our implementation, we utilize a variant KL algorithm. With a random initialized network partition $\{B^{(k)}\}_{k=1}^{K}|_{t=0}$ for $\mathcal{M}$ at $t = 0$, we iteratively find the optimal separation by swapping nodes (network layer). Given the two consecutive block $B^{(k)}|_t = F_i^{(l)} \cdots \circ F_i^{(l+p_k)}$ and

$B^{(k+1)}|_t = F_i^{(l+p_k+1)} \cdots \circ F_i^{(l+p_k+p_{k+1})}$ at time $t$, we conduct a forward and a backward neural network layer swap between successive blocks, whereas the partition achieving the largest objective value becomes the new partition

$$(B^{(k)}|_{t+1}, B^{(k+1)}|_{t+1}) = \arg\max\{J(B^{(k)}|_t, B_i^{(k+1)}|_t), J(B_i^{(k)}|_t^{\text{f}}, B_i^{(k+1)}|_t^{\text{f}}), J(B_i^{(k)}|_t^{\text{b}}, B_i^{(k+1)}|_t^{\text{b}})\} \tag{6}$$

where $(B_i^{(k)}|_t^{\text{f}}, B_i^{(k+1)}|_t^{\text{f}}) = B^{(k)}|_t \xrightarrow{F_i^{l+p_k}} B_i^{(k+1)}, (B_i^{(k)}|_t^{\text{b}}, B_i^{(k+1)}|_t^{\text{b}}) = B^{(k)}|_t \xleftarrow{F_i^{l+p_k+1}} B_i^{(k+1)}$ (7)

For the outer optimization, we do a K-Means [52] style clustering. With the current network partition $\{B^{(k)*}\}_{k=1}^K$, we alternate between assigning each block to a equivalence set $G_j$, and identifying the anchor block within each set $B_{a_j} \in G_j$. It has been proved that both KL and K-Means algorithms converge to a local minimum according to the initial partition and anchor selection. We repeat the optimization for $R = 200$ runs with different seeds and select the best partition as our final results.

### 3.3 Network Reassembly by Solving an Integer Program

As we have divided each deep network into $K$ partitions, each belongs to one of the $K$ equivalence sets, all we want now is to find the best combination of neural blocks as a new pre-trained model under certain computational constraints. Consider $K$ disjoint equivalence sets $G_1, \ldots, G_K$ of blocks to be reassembled into a new deep network of parameter constraint $C_{\text{param}}$ and computational constraint $C_{\text{FLOPs}}$, the objective is to choose exactly one block from each group $G_j$ as well as from each network stage index $j$ such that the reassembled model achieves optimal performance on the target task without exceeding the capacity. We introduce two the binary matrices $X_{(ik,j)}$ and $Y_{(ik,j)}$ to uniquely identity the reassembled model $\mathcal{M}(X, Y)$. $X_{(ik,j)}$ takes on value 1 if and only if $B_i^{(k)}$ is chosen in group $G_j$, and $Y_{(ik,j)} = 1$ if $B_i^{(k)}$ comes from the $k$-th block. The selected blocks are arranged by the block stage index. The problem is formulated as

$$\max_{X,Y} P_T(\mathcal{M}(X, Y)) \tag{8}$$

$$\text{s.t. } |\mathcal{M}(X, Y)| \leq C_{\text{param}}, FLOPs(\mathcal{M}(X, Y)) \leq C_{\text{FLOPs}} \tag{9}$$

$$\sum_{i=1}^N \sum_{k=1}^K X_{(ik,j)} = 1, X_{(ik,j)} \in \{0, 1\}, j = 1, \ldots, K \tag{10}$$

$$\sum_{i=1}^N \sum_{j=1}^K Y_{(ik,j)} = 1, Y_{(ik,j)} \in \{0, 1\}, k = 1, \ldots, K \tag{11}$$

where $P_T$ is again the task performance. Equation 10 and 11 indicates that each model only possesses a single block from each equivalence set and each stage index. As such, the reassembled blocks are automatically ordered through its stage index in their original model. The problem falls exactly into a *0-1 Integer Programming* [60] problem with a non-linear objective. Conventional methods train each $\mathcal{M}(X, Y)$ to obtain $P_T$. Instead of training each candidate till convergence, we estimate the transfer-ability of a network by counting the linear regions in the network as a training-free proxy.

**Estimating the Performance with Training-Free Proxy.** The number of linear region [56, 26] is a theoretical-grounded tool to describe the expressivity of a neural network, which has been successfully applied on NAS without training [55, 10]. We, therefore, calculate the data-dependent linear region to estimate the transfer performance of each model-task combination. The intuition is straightforward: the network can hardly learn to distinguish inputs with similar binary codes.

We apply random search to get a generation of reassembly candidates. For a whole mini-batch of inputs, we feed them into each network and binarize the features vectors using a sign function. Similar to NASWOT [55], we compute the kernel matrix $\mathbf{K}$ using Hamming distance $d(\cdot, \cdot)$ and rank the models using $\log(\det \mathbf{K})$. Since the computation of $\mathbf{K}$ requires nothing more than a few batches of network forwarding, we replace $P_T$ in Equation 8 with NASWOT score for fast model evaluation.

## 4 Experiments

In this section, we first explore some basic properties of the the proposed DeRy task, and then evaluate our solution on a series of transfer learning benchmarks to verify its efficiency.

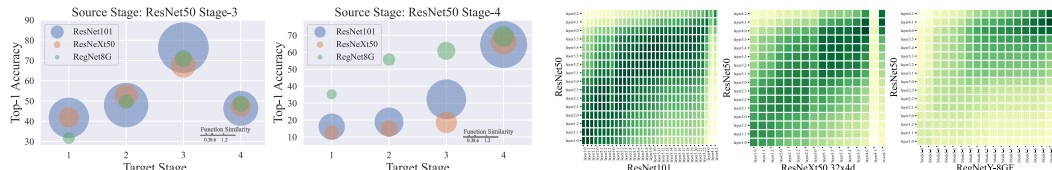

**Figure 4:** FROZEN-TUNING accuracy on ImageNet by replacing the 3rd and 4th stage of R50 to target blocks.

**Figure 5:** Pair-wise Linear CKA between pre-trained R50 and (1) R101 (2) RX50 and (3) Reg8G.

**Model Zoo Setup.** We construct our model zoo by collecting pre-trained weights from Torchvision [1], timm [2] and OpenMMlab [3]. We includes a series of manually designed CNN models like ResNet [30] and ResNeXt[84], as well as NAS-based architectures like RegNetY [63] and MobileNetv3 [34]. Due to recent popularity of vision transformer, we also take several well-known attention-based architectures into consideration, including Vision Transformer (ViT) [20] and Swin-Transformer [49]. In addition to the differentiation of the network structure pre-trained on ImageNet, we include models with a variety of pre-trained strategies, including SimCLR [8], MoCov2 [11] and BYOL [24] for ResNet50, MoCov3 [12] and MAE [29] for ViT-B. Those models are pre-trained on ImageNet1k [68], ImageNet21K [67], Xrays [15] and iNaturalist2021 [77], Finally we result in 21 network architectures, with 30 pre-trained weights in total. We manually identify the atomic node to satisfy our line graph assumption. Each network is therefore a line graph composed of atomic nodes.

**Implementation details.** For all experiments, we set the partition number $K = 4$ and the block size coefficient $\epsilon = 0.2$. We sample 1/20 samples from each train set to calculate the linear CKA representation similarity. The NASWOT [55] score is estimated with 5-batch average, where each mini-batch contains 32 samples. We set 5 levels of computational constraints, with $C_{\text{param}} \in \{10, 20, 30, 50, 90\}$ and $C_{\text{FLOPs}} \in \{3, 5, 6, 10, 20\}$, which is denoted as DeRy($K$, $C_{\text{param}}$, $C_{\text{FLOPs}}$). For each setting, we randomly generated 500 candidates. Each reassembled model is evaluate under 2 protocols (1) FROZEN-TUNING. We freeze all trained blocks and only update the parameter for the stitching layer and the last linear classifier and (2) FULL-TURNING. All network parameter are updated. All experiments are conducted on a $8 \times$ GeForce RTX 3090 server. To reduce the feature similarity calculation cost, we construct the similarity table *offline* on ImageNet. The complexity analysis and full derivation are shown in the Appendix.

### 4.1 Exploring the Properties for Deep Reassembly

**Similarity, Position and Reassembly-ability.** Figure 4 validates our functional similarity, reassembled block selection, and its effect on the model performance. For the ResNet50 trained on ImageNet, we replace its 3rd and 4th stage with a target block from another pre-trained network (ResNet101, ResNeXt50 and RegNetY8G), connected by a single stitching layer. Then, the reassembled networks are re-trained on ImageNet for a 20 epochs under FROZEN-TURNING protocol. The derived functional similarity in Section 3.2 is shown as the diameter of each circle. **We observe that**, the stitching position makes a substantial difference regarding the reassembled model performance. When replaced with a target block with the same stage index, the reassembled model performs surprisingly well, with $\geq 70\%$ top-1 accuracy, even if its predecessors are trained with different architectures, seeds, and hyperparameters. **It is also noted** that, though function similarity is not numerically proportional to the target performance, it correctly reflects the performance ranking within the same target network. It suggests that our function similarity provides a reasonable criteria to identify equivalence set. In sum, the coupling between the *similarity-position-performance* explains our design to select one block from each equivalence set as well as the stage index. We also visualize the linear CKA [40] similarity between the R50 and the target networks in Figure 5. An interesting finding is that *diagonal pattern* for the feature similarity. The representation at the same stage is highly similar. More similarity visualizations are provided in the Appendix.

**Partition Results.** Due to the space limitation, the partition results of the model zoo are provided in the Appendix. Our observation is that, the equivalent sets tend to *cluster the blocks by stage index*.

---

[1]https://pytorch.org/vision/stable/index.html

[2]https://github.com/rwightman/pytorch-image-models

[3]https://github.com/open-mmlab

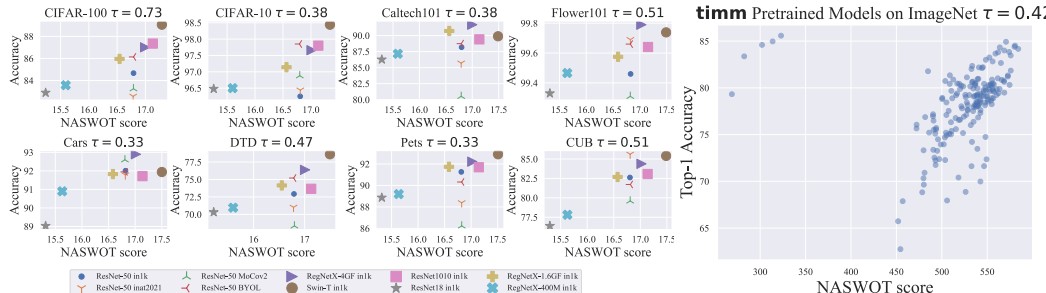

**Figure 6:** Plots of NASWOT [55] score and test accuracy for (Left) 10 pre-trained model on 8 downstream tasks and (Right) `timm` model zoo on ImageNet. $\tau$ is the Kendall's Tau correlation.

For example, all bottom layers of varied pre-trained networks are within the same equivalence set. It provides valuable insight that neural networks learns similar patterns at similar network stage.

**Architecture or Pre-trained Weight.** Since `DeRy` searches for the architecture and weights concurrently, a natural question arises that "Do both *architecture* and *pre-trained weights* lead to the final improvement? Or only *architecture* counts?" We provide the experiments in the Appendix that both factors contribute. It is observed that training the `DeRy` architecture from scratch leads to a substantial performance drop compared with `DeRy` model with both new structures and pre-trained weights. It validates our arguments that our reassembled models benefit from the pre-trained models for efficient transfer learning.

**Verifying the training-free proxy.** As the first attempt to apply the NASWOT to measure model transfer-ability, we verify its efficacy before applying it to `DeRy` task. We adopt the score to rank 10 pre-trained models on 8 image classification tasks, as well as the `timm` model zoo on ImageNet, shown in Figure 6. We also compute the Kendall's Tau correlation [37] between the fine-tuned accuracy and the NASWOT score. It is observed that the NASWOT score provides a reasonable predictor for model transfer-ability with a high Kendall's Tau correlation.

## 4.2 Transfer learning with Reassembled Model

**Evaluation on ImageNet1k.** We first compare the reassembled network on ImageNet [68] with current best-performed architectures. We train each model for either 100 epochs as SHORT-TRAINING or a 300 epochs as FULL-TRAINING. Except for `DeRy`, all models are trained from scratch. We optimize each network with AdamW [50] alongside a initial learning rate of $1e-3$ and cosine lr-decay, mini-batch of 1024 and weight decay of 0.05. We apply RandAug [17], Mixup [95] and CutMix [93] as data augmentation. All model are trained and tested on $224$ image resolutions.

Table 7 provides the Top-1 accuracy comparison on Imagenet with various computational constraint. We underline the best-performed model in the model zoo. **First**, It is worth-noting that `DeRy` provide very competitive model, even under FROZEN-TURNING or SHORT-TRAINING protocol. `DeRy(4,90,20)` manages to reach $78.6\%$ with 1.27M parameter trainable, which provides convincing clue that the heterogeneous trained model are largely graftable. With only SHORT-TRAINING, `DeRy` models also match up with the full-time trained model in the zoo. For example, `DeRy(4,10,3)` gets to $76.9\%$ accuracy within 100 epochs' training, surpassing all small-sized models. The performance can be further improved towards $78.4\%$ with the standard 300-epoch training. **Second**, `DeRy` brings about faster convergence. We compare with ResNet-50 and Swin-T under the same SHORT-TRAINING setting in Table 9 and Figure 10. It is clear that, by assembling the off-the-self pre-trained blocks, the `DeRy` models can be optimized faster than the it competitors, achieving 0.9% and 0.2% accuracy improvement over the Swin-T model with less parameter and computational requirements. **Third**, as showcased in Figure 8, our `DeRy` is able to search for diverse and hybrid network structures. `DeRy(4,10,3)` learns to adopt light-weight blocks like MobileNetv3, while `DeRy(4,90,20)` gets to a large CNN-Swin hybrid architecture. Similar hybrid strategy has been proved to be efficient in manual network design [54, 83].

**Transfer Image classification.** We evaluate transfer learning performance on 9 natural image datasets. These datasets covered a wide range of image classification tasks, including 3 object classification

| Architecture | #Train/All Params (M) | FLOPs (G) | Top-1 |
|---|---|---|---|
| RSB-ResNet-18 | 11.69/11.69 | 1.82 | 70.6 |
| RegNetY-800M | 6.30/6.30 | 0.8 | 76.3 |
| ViT-T16 | 5.7/5.7 | 1.3 | 74.1 |
| DeRy(4,10,3)-FZ[†] | 1.02/7.83 | 2.99 | 41.2 |
| DeRy(4,10,3)-FT[†] | 7.83/7.83 | 2.99 | 76.9 |
| DeRy(4,10,3)-FT | 7.83/7.83 | 2.99 | 78.4 |
| RSB-ResNet-50 | 25.56/25.56 | 4.12 | 79.8 |
| RegNetY-4GF | 20.60/20.60 | 4.0 | 79.4 |
| ViT-S16 | 22.0/22.0 | 4.6 | 79.6 |
| Swin-T | 28.29/28.29 | 4.36 | 81.2 |
| DeRy(4,30,6)-FZ[†] | 1.57/24.89 | 4.47 | 60.5 |
| DeRy(4,30,6)-FT[†] | 24.89/24.89 | 4.47 | 79.6 |
| DeRy(4,30,6)-FT | 24.89/24.89 | 4.47 | 81.2 |
| RSB-ResNet-101 | 44.55/44.55 | 7.85 | 81.3 |
| RegNetY-8GF | 39.20/39.20 | 8.1 | 81.7 |
| Swin-S | 49.61/49.61 | 8.52 | 82.8 |
| DeRy(4,50,10)-FZ[†] | 3.92/40.41 | 6.43 | 72.0 |
| DeRy(4,50,10)-FT[†] | 40.41/40.41 | 6.43 | 81.3 |
| DeRy(4,50,10)-FT | 40.41/40.41 | 6.43 | 82.3 |
| RegNetY-16GF | 83.6/83.6 | 16.0 | 82.9 |
| ViT-B16 | 86.86/86.86 | 33.03 | 79.8 |
| Swin-B | 87.77/87.77 | 15.14 | 83.1 |
| DeRy(4,90,20)-FZ[†] | 1.27/80.66 | 13.29 | 78.6 |
| DeRy(4,90,20)-FT[†] | 80.66/80.66 | 13.29 | 82.4 |
| DeRy(4,90,20)-FT | 80.66/80.66 | 13.29 | 83.2 |

**Table 7:** Top-1 accuracy of models trained on ImageNet. † means the model is trained for 100 epochs. "FZ" and "FT" denote the reassembled blocks are frozen or fine-tuned. Trainable parameters are marked in red.

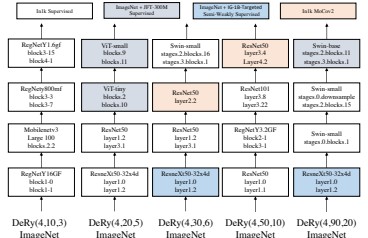

**Figure 8:** Reassembled structures on ImageNet.

| Architecture | Params (M) | FLOPs (G) | Top-1 | Top-5 |
|---|---|---|---|---|
| ResNet-50 | 25.56 | 4.12 | 76.8 | 93.3 |
| Swin-T | 28.29 | 4.36 | 78.3 | 94.6 |
| DeRy(30, 6)-FT | **24.89** | 4.47 | **79.6** | **94.8** |
| ResNet-101 | 44.55 | 7.85 | 79.0 | 94.5 |
| Swin-S | 49.61 | 8.52 | 80.8 | **95.7** |
| DeRy(50, 10)-FT | **40.41** | **6.43** | **81.2** | 95.6 |

**Table 9:** Top-1 and Top-5 Accuracy for the ImageNet 100-epoch FULL-TUNING experiment.

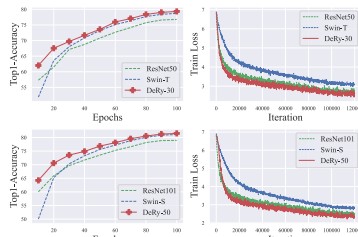

**Table 10:** (Left) Test accuracy and (Right) Train loss comparison under the 100-epoch training on ImageNet.

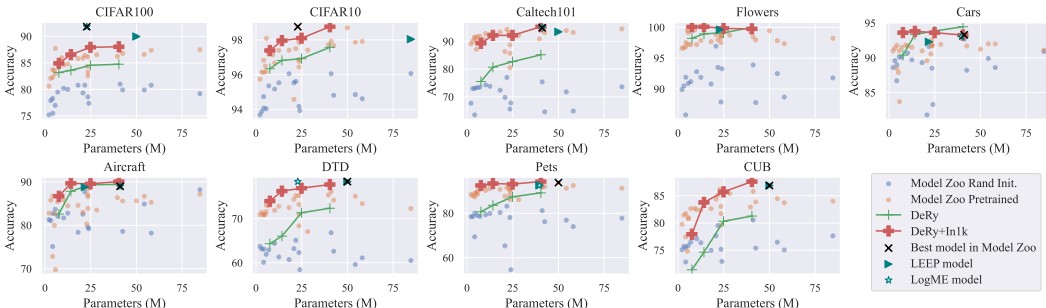

**Figure 11:** Transfer performance on 9 image classification tasks with the model zoo and our DeRy. Each blue or orange point refers to a single model trained from scratch or pre-trained weights.

tasks CIFAR-10 [43], CIFAR-100 [43] and Caltech-101 [22]; 5 fine-grained classification tasks Flower-102 [59], Stanford Cars [42], FGVC Aircraft[53], Oxford-IIIT Pets [61] and CUB-Bird [79] and 1 texture classification task DTD [14]. We FULL-TUNE all candidate networks in the model zoo and compare them with our DeRy model. Two model selection strategies LogME [92] and LEEP [57] are also taken as our baselines. For fair comparison, we further train the reassembled network on ImageNet for 100 epochs to further boost the transfer performance. Following [8, 41], we perform hyperparameter tuning for each model-task combination, which are elaborated in the Appendix.

Figure 11 compares the transfer performance between our proposed DeRy and all candidate models. By constructing models from building blocks, the DeRy generally surpasses all network trained from scratch within the same computational constraints, even beats pre-trained ones on Cars, Aircraft, and Flower. If allowing for pre-training on ImageNet (DeRy+In1k), we can further promote the test accuracy, even better than the best-performing candidate in the original model zoo (highlighted by ×). The performance improvement rises up as parameter constraints increase, which demonstrates the scalability of the proposed solution. Model selection approaches like LogME and LEEP may not necessarily get the optimal model, thus failing to release the full potential of the model zoo. These

findings provide encouraging evidence that DeRy gives rise to an alternative approach to improve the performance when transferring from a zoo of models.

## 4.3 Ablation Study

To support the effectiveness of the DeRy pipeline, we further verify influence of each design in our solution through ablation studies.

**Partition and Reassembly Strategy.** First, we conduct experiments by replacing the (1) cover set partition and (2) training-free reassembly with random partition or search. For the partition ablation, we randomly dissect each network into $K$ partitions and reassemble the blocks in an order-less manner using our training-free proxy. For the reassembly ablation, we retain the cover set partition and fine-tune each randomly reassembled network for 100 epochs. Due to the computation limitation, we can only evaluate

| Cover Set Partition | Train-Free Reassembly | Acc (%) | Search Cost (GPU days) |
|---|---|---|---|
| ✓ | ✓ | 72.0 | 0.23 |
| ✗ | ✓ | 70.5 | 1.48 |
| ✓ | ✗ | 73.5 | 135 |
| ✗ | ✗ | 72.2 | 135 |

**Table 2:** Ablation study on partition and reassembly strategy.

25 candidates for reassembly ablation. We report the 100-epoch FROZEN-TUNING top-1 accuracy and the search time on ImageNet in Table 2 under the DeRy(4,50,10) setting. Note that we do not include the similarity computation time into our account since it is computed *offline*. We see that the majority of the search cost comes from the fine-tuning stage. The training-free proxy largely alleviates the tremendous computational cost by $10^4$ times, with marginal performance degradation. On the other hand, the cover set model partition not only improves the transfer performance but also reduces the reassembly search space from $O(\prod_{i=1}^{N} \binom{L_i-1}{K-1})$ to $O(1)$. Both stages are crucial.

**Granularity of Partition.** To see the impact of partition number $K$, here we show the experimental results with different $K \in \{4, 5, 6\}$. We set the configuration to DeRy(K, 30, 6). The reassembled network is trained with FULL-TUNING setting on ImageNet for 100 epochs. We report the parameter size, FLOPs as well as their top-1 and top-5 accuracy. As demonstrated in the Table 3, We notice that, as the partition number $K$ increases, the performance of the reassembled model remains quite stable or slightly increase. It suggest that DeRy is highly flexible with different granularity of partitioning.

| Partition Number | # Param | FLOPs | Top-1 | Top-5 |
|---|---|---|---|---|
| $K = 4$ | 24.89 | 4.47 | 79.6 | 94.8 |
| $K = 5$ | 21.14 | 5.53 | 79.7 | 94.9 |
| $K = 6$ | 23.38 | 5.39 | 79.8 | 95.0 |

**Table 3:** Ablation study on partition granularity with $K \in \{4, 5, 6\}$.

## 5 Conclusion

In this study, we explore a novel knowledge-transfer task called Deep Model Reassembly (DeRy). DeRy seeks to deconstruct heterogeneous pre-trained neural networks into building blocks and then reassemble them into models subject to user-defined constraints. We provide a proof-of-concept solution to show that DeRy can be made not only possible but practically efficient. Specifically, pre-trained networks are partitioned jointly via a cover set optimization to form a series of equivalence sets. The learned equivalence sets enable choosing and assembling blocks to customize networks, which is accomplished by solving integer program with a training-free task-performance proxy. DeRy not only achieves gratifying performance on a series of transfer learning benchmarks, but also sheds light on the functional similarity between neural networks by stitching heterogeneous models.

**Acknowledgement**

This research is supported by the National Research Foundation, Singapore under its AI Singapore Programme (AISG Award No: AISG2-RP-2021-023). Xinchao Wang is the corresponding author.

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
