# OpenReview forum: "Deep Model Reassembly"
_NeurIPS.cc/2022/Conference — NeurIPS 2022 Accept_

### Official Review · Reviewer_q2yK · 2022-07-10

**Rating:** 6
**Confidence:** 4
**Soundness:** 3 good
**Presentation:** 3 good
**Contribution:** 3 good

**Summary:**

The paper proposes to reuse the building blocks of pretrained neural networks for new tasks by resembling them under given computation constraints. The proposed approach, DeRy, first learns to partition the layers of the base networks jointly into equivalent sets via a cover set optimization, then selects and stitchs the optimal blocks into a new network by solving an integer programming problem. DeRy is evaluated on ImageNet1K and 9 other transfer-learning benchmarks via linear probing and finetuning.

**Questions:**

i) What is used as the s(*,*) metric. From the paper, I would guess it is linear CKA [33]?

ii) The overhead (complexity) of building the s(*,*) index, partition, and stitching. Table 2 shows the GPU time. I wonder if it includes building the s(*,*) index.

iii) I wonder if the s(*,*) table needs to be re-built for each dataset? If not, I wonder what is the main dataset used for computing the s(*,*) table (e.g. the GPU time in Table 2 is for what dataset)

iv) I wonder if the partition and stitching need to be re-run for each dataset? (e.g. if the DeRy models in Table 7 and Fig 11 are the same).

v) the list of the 28 base models used

vi) It seems Fig. 11 only shows a subset of the models (less than 28 models). I wonder what are the models shown in Fig. 11.

vii) the detailed architectures of the obtained DeRy models


**Limitations:**

The limitations and the potential negative societal impact are NOT discussed in the submission.

**Strengths And Weaknesses:**

Strengths

i) the paper introduces a new task, deep model reassembly. The task is ambitious!

ii) it is interesting to learn that the task, when defined under specific assumptions, has a feasible solution that works to a certain extent.

iii) the writing is clear in general


Weaknesses

The formulation is not as general as I would expect. I have concerns about the practical usefulness of the method and whether it could scale.

i) The model space is small. DeRy assumes the base model forms a path graph (L170-171), which excludes architectures with heavy skip connections, e.g. UNets. The granularity of the partition is small, i.e. K = 4. The number of complexity levels, i.e. 5 param/gflop levels,  is also small, which excludes efficient model architectures (e.g. < 1 gflops) that are of great interest to the community.

ii) Performance: on ImageNet1k, DeRy shows no advantage. On the transfer-learning datasets, DeRy shows marginal advantages with further pretraining on ImageNet1k. From L311-312, the hyperparameters for DeRy were tuned for each model-task combination, I wonder if the hyperparameters for the base models were also tuned in this way?

iii) Overhead. I would guess building the s(*,*) table involves computation of R * (N * N * K * K)/20 epochs (from L237, 1/20 training samples are used). Even with a small granularity (N=28, K=4, R=200), the computation looks huge?

iv) Insights:  from Fig.1 in the suppl. , the partition tends to group blocks of the same stage id, with the current K (i.e. K=4), it seems the partition algorithm obtains a trivial solution? If this is the case, do we need the partition step at all?



Minor issues

i) Typos, L73 trasnfer → transfer L110 (k) → (l), L170 graphs → graph, L248 ImagNet → ImageNet, L308 testure –> texture, L89 (suppl): #praram → #param
ii) L121 “It is clear that no single model universally dominants in transfer evaluations. It builds up our primary motivation to reassemble trained models rather than trust the “best” candidate. ” Arguably, DeRy does not provide a universally dominant model either.

---

> ### Author Response · Authors · 2022-08-02
> **Response to Reviewer q2yK (Part Ⅲ)**
>
> `>>> Q7` **Trivial Solution of Partition Algorithm and Its Necessity**
>
> `>>> A7` Thanks for the comment. Our partition results are indeed not trivial solutions. We start the partition by treating all the blocks equally without any prior knowledge, bias, or assumptions, and then carry out the proposed representation-based clustering. It turns out that, encouragingly, the obtained clusters mostly align with our human intuitions, where network blocks at similar depths have similar functionality.
>
> In other words, we would consider this coincidence as an interesting observation that follows our intuitive hypothesis, which, without the proposed partition, cannot be validated. In fact, through our experiment, we did observe some exceptional cases where the network blocks are not grouped by the stage ID, indicating the necessity of network partition.
>
> `>>> Q8` **Minor issues**
>
> `>>> A8` We truly appreciate the reviewer for the comment. We have corrected the typo and uploaded the new version for the reviewer's reference.
>
>
>
> `>>> Q9` **i) What is used as the s(,) metric?**
>
> `>>> A9` Yes, we use Linear CKA for representation calculation. It has been mentioned in `Line 238-239`.
>
> `>>> Q10` **ii) The overhead (complexity) of building the s(,) index, partition, and stitching. Table 2 shows the GPU time. I wonder if it includes building the s(,) index.**
>
> `>>> A10` We thank the reviewer for the comment. The computation for building the s(,) index is described above. The partition computation is fast, with around 5 seconds per run and we run 200 times to ensure convergence. For the reassembly stage,  we need to evaluate 500 candidates, each with a 5-batch average to estimate the NASWOT score and.  Total stitching time is less than 5 hours. We do not include the similarity calculation time in `Table 2`, since it is a pre-processing step. We only account for the partition and reassembly search time. Even if we include it in `Table 2`, only 1 GPU day is accumulated to the first row, which is still faster and more efficient compared to Row 2 without partition.
>
>
> `>>> Q11` **iii) I wonder if the s(,) table needs to be re-built for each dataset?**
>
> `>>> A11` Representation similarity s(,) is only built on ImageNet and then it is kept fixed for all downstream datasets. As mentioned above, 1 GPU day is needed to compute a representation similarity table.
>
> `>>> Q12` **iv) I wonder if the partition and stitching need to be re-run for each dataset?**
>
> `>>> A12` No, we do not re-run partitions and stitching on each dataset. In our paper, we run the DeRy on ImageNet and apply it to all downstream tasks, which saves the cumbersome search cost on every task. In fact, this is the common practice for other tasks, such as NAS, to search on proxy data and then train on target ones.
>
> `>>> Q13` **v) The list of the 28 base models used**
>
> `>>> A13` Thanks for the comment. In total, there are 21 architectures and 28 pre-trained weights.
>
> 1.  Swin-T/S/B/L sup in1k(4), Swin-B sup in21k(1),
>
> 2.  ResNet18/50/101 sup in1k(3), ResNet50 MoCov2/SimCLR/BYOL in1k(3), ResNet50 sup iNatualist (1),
>
> 3.  RegNetY-800MF/1.6GF/3.2GF/8GF/16GF/32GF sup in1k(6),
>
> 4.  ViT-T/S/B/L sup in1k(4), ViT-S MoCov3 in1k(1), ViT-B MAE in1k(1),
>
> 5.  MobilenetV3 Large 1.0/0.75 sup in1k(2),
>
> 6.  ResNeXt 50 32x4d/ResNeXt 101 32x8d sup in1k(2).
>
> Please see `Table 1 and Table 2 of the Supplementary` and code `blocklize/block_meta.py` for more details.
>
> `>>> Q14`  **vi) Models shown in Fig. 11**
>
> `>>> A14` Thanks for the comment. We would like to include as many models as possible in `Fig 11`; however, some of the models are excluded for better visualization quality. Instead, all the detailed results are shown in `Table 1 and Table 2 of the Supplementary`. Specifically, we exclude extra-large (e.g. ViT-L, Swin-L) or models with extremely poor performance (e.g. Swin-T only gets an accuracy of 5.3% on Flower) because they are far from the majority of data points.
>
> `>>> Q15`  **vii) The detailed architectures of the obtained DeRy models.**
>
> `>>> A15` Thanks for the comment. Since we only search once on ImageNet, the detailed architecture is already mentioned in `Figure 8`. The stitching layer structure is also provided in `Supplementary Table 3`.

---

> > ### Author Response · Authors · 2022-08-08
> > **Thank the Reviewer for the Constructive Comments**
> >
> > Dear Reviewer,
> >
> > We would like to thank you again for your constructive comments and kind effort in reviewing our submission. Please do let us know if our response has addressed your concerns, and we are more than happy to address any further comments.
> >
> > Thanks!

---

> > > ### Comment · Reviewer_q2yK · 2022-08-09
> > > **Comment after the rebuttal**
> > >
> > > Thank you for the detailed response that resolves most of my concerns. As the first effort toward reusing/resembling pretrained models, the paper presents promising results. I’d like to raise the score to weak accept (6). The authors are encouraged to include the rebuttal in the final paper to provide more context about the task and the solution. From the discussion, there seem to be open problems down the road, which could be explored in future research.

---

> ### Author Response · Authors · 2022-08-02
> **Response to Reviewer q2yK (Part Ⅱ)**
>
> `>>> Q3` **Complexity Level and Efficient Model Architectures**
>
> `>>> A3` Thanks for the comment. To our best knowledge, 5 param/gflop levels are indeed sufficient as compared to prior works for neural architecture designs. For example, ViT has 5 scales of ViT-T/S/B/L/H, ResNet has 5 scales of ResNet18/34/50/101/152, MobileNetv3 has 4 scales of V3-Large 1.0/0.75, and V3-Small 1.0/0.75. The current 5 computational scales, therefore, indeed cover most of the models in the model zoo.
>
> As suggested by the reviewer, we add a new configuration of `DeRy(4, 8, 1)` to demonstrate DeRy’s capability to reassemble efficient network architectures. Specifically, we train the network on ImageNet for 300 epochs and compare it with other efficient models. The parameter number/GFLOPs and top-1 accuracy on ImageNet are provided in the table below. We note that our DeRy still delivers competitive results with only  `< 1 gflops` computational overhead.
> | Network                | \# Param  | GFLOPs    | ImageNet Top1 Acc |
> | ---------------------- | --------- | --------- | ----------------- |
> | ViT-T16                | 5.7       | 1.3       | 74.1              |
> | MobileNet v3 Large 1.0 | 5.4       | 0.23      | 74.0             |
> | RegNetY-600M           | 6.1       | 0.6       | 75.5              |
> | EFFICIENTNET-B1  | 7.8 | 0.7 | 75.9              |
> | DeRy(4, 8, 1)          | 6.19      | 0.86      | **76.0**              |
>
> `>>> Q4`**Performance on ImageNet and Transfer-learning Datasets**
>
> `>>> A4`
> 1.  **On ImageNet**, DeRy shows on-par or superior performances but comes with a much faster convergence speed. For example, under DeRy(4,10,3), our small model yields a result of 2% improvement as compared to counterparts with comparable complexity.
>
> 	Also, as shown in Table 7, DeRy under 100-epoch training is comparable to counterpart networks trained for 300 epochs; on the other hand, when both are trained for 100 epochs, DeRy delivers results better than ResNet and Swin (Table 9 and Fig 10). As such, DeRy indeed yields significantly finer results when performance and convergence combined are considered.
>
> 2.  **On 8 transfer learning datasets**,  DeRy, without pretraining, consistently outperforms all the random initialized nets, in many by a significant gap, as shown in `Fig 11`. In the case of Flower and Cars with configurations `DeRy(4,10,3)` and `DeRy(4,30,6)`, for example, the improvement is around 1~2 %, which is truly not marginal. The exact performances can be found in Table 1 in Supplementary Material.
>
>
> `>>> Q5` **Were the hyper-parameters for the base models also tuned in this way?**
>
> `>>> A5` Yes. As described in `Line 314-315`, we perform careful parameter tuning for all model-task combinations, not only our DeRy model. The hyper-parameter search space is described in `Supplementary Material Line 164-184`.
>
> `>>> Q6` **Computation for Similarity**
>
> `>>> A6`Thanks for the comment. Indeed, the similarity computation is heavy, as discussed in `Section 2.4 of the supplementary material`. However, in practice, this estimation can be pre-computed offline and saved to a lookup table, which, in turn, greatly reduces the computation time. In fact, this gives us a **speed up of more than 20 times** as compared to the computational overhead listed by the reviewer, which applies to online computation.
>
> Specifically, we may repeat the similarity calculation for the same pair of networks over iterations. In our implementation, all similarity tables are built before partitioning using an offline manner. Hence, the computation of `s(,)` is independent of $K$ and $R$. Assume that, for $B$ batches of data, we pass it through each of the $N$ networks once for $N \times B$ times, and collect all intermediate features and save them into local files. This process is fast, which takes no more than 20 mins on an 8x3090 server for all 28 models. Next, we conduct `s(,)` computation for $\sum_{i, j=1,\dots, N} L_i * L_j$ times for each pair of layers. This is where the large computation overhead comes from since there is a large matrix multiplication for each `s(,)`. We implement a mini-batch version full CKA algorithm to save memory and do multi-thresholding. As such, the computation for each pair of networks reduces to around 1~2 min. For a zoo of 28 models, we need around $28\times 28 \times 1 \sim 28\times 28 \times 2$ minutes, with about 12-24 hour in total. Once the similarity computation is done, the partition and reassembly steps can be done without hassle. Please see our code at `similarity/get_rep.py` and `similarity/compute_sim.py` on the implementation details.
>
> Notably, this large computational overhead is attributed to estimating representation similarity itself, which is in fact orthogonal to the proposed DeRy pipeline. When other efficient representation-similarity estimations are available, we may readily adopt them to DeRy and accelerate the overall process.

---

> ### Author Response · Authors · 2022-08-02
> **Response to Reviewer q2yK (Part I)**
>
> We would like to thank the reviewer for the insightful feedback and comments. We are encouraged that the reviewer found the paper well written and the problem setup is interesting and ambitious. We address the reviewer’s comments below and will include all the feedback in the revised version of the manuscript.
>
> `>>> Q1` **Path Graph**
>
> `>>> A1` Admittedly, as the first endeavor towards general network reassembly, we focus on path-graph-based models for illustrating our strategy; heavy skip-based architectures are, in fact, *intentionally* excluded from this pilot study, so as to make this investigation more self-contained and logically clear.
>
> Nevertheless, we are not the only ones who made such assumptions. The path-based hypothesis is indeed prevalent for pilot studies among other research tasks: for cell-based neural architecture search[A], network stitching[12], and gaussian process understanding of DNN[B], pioneering attempts all rely on the assumption of path-graph models, based on which variants and extensions thrive in the following work.
>
> In fact, DeRy can be extended to network architectures with skip connections by applying node duplication and introducing a novel feature similarity estimation: node duplication aims to remove the skip connections of the original graph topology, while the similarity estimation facilitates the network partition. For example, given a four-layer UNet $A$ with architecture `L1->L2->L3->L4` and skip-connection `L1->L4`, we can replicate node `L1` into two identical layers `L’1` and `L’’1`; this results in a transformed graph with two paths `L’1->L2->L3->L4` and `L’’1->L4`, where `L4` is the joint node. As a result, the transformed network has no skip-connections but a multi-branch structure. For any subgraph $A’$ from $A$ with a line graph structure, we can conduct the network partition as described in our manuscript. For subgraphs $A’$ with multiple inputs or multiple outputs, we may define a new multiple-to-multiple representation similarity `s*()` to carry out the partition. Specifically, `s*()` takes two sets of features and produces a similarity score. The original one-to-one representation similarity is then replaced by  `s*()` to further cluster and partition the network. As such, we can assemble the derived block with any graph topology. We would like to provide a more comprehensive study on this in our future work.
>
> [A] Liu H, Simonyan K, Yang Y. Darts: Differentiable architecture search[J]. arXiv preprint arXiv:1806.09055, 2018.
>
> [B] Lee J, Bahri Y, Novak R, et al. Deep neural networks as gaussian processes[J]. arXiv preprint arXiv:1711.00165, 2017.
>
> `>>> Q2` **Granularity of Partition K = 4**
>
> `>>> A2` We thank the reviewer for the comment. In fact, the granularity of partition `K` cannot be set to be too large, since small networks like ResNet18 contain only 8 nodes, while ViT-T/S/B contains only 12 nodes. As such, `K=4` is indeed a reasonable choice, as the number of nodes in various networks is mostly multipliers of 4, and most manual-designed networks have 4 stages.
>
> To see the impact of `K`, here we show the experimental results with different `K` settings, i.e., `K=4/5/6`, in the table below. We set the configuration to `DeRy(K, 30, 6)`. The reassembled network is trained on ImageNet for 100 epochs. All other experimental settings are kept the same as those in the manuscript. We find that, as the partition number `K` increases, the performance of the reassembled model remains quite stable.
>
> | K | \# Param | GFLOPs | Top1 Acc | Top5 Acc |
> | - | -------- | ------ | ----------------- | ----------------- |
> | 4 | 24.89    | 4.47   | 79.62              | 94.83             |
> | 5 | 21.14    | 5.53   | 79.68             | 94.89             |
> | 6 | 23.38    | 5.39   | 79.83             | 95.02             |

---

### Official Review · Reviewer_3qgQ · 2022-07-11

**Rating:** 8
**Confidence:** 5
**Soundness:** 3 good
**Presentation:** 4 excellent
**Contribution:** 4 excellent

**Summary:**

This paper presents a novel knowledge-transfer task, termed as Deep Model Reassembly (DeRy), for general-purpose model reuse. Specifically, given a collection of heterogeneous models pre-trained from distinct sources and with diverse architectures, authors first dissect each model into distinctive building blocks, and then selectively reassemble the derived blocks to produce customized networks under both the hardware resource and performance constraints. Experiments on ImageNet dataset show the promising performance of the proposed method under different settings.

**Questions:**

Address weaknesses. I would like the authors to address my questions regarding experiments and comparisons as described above.

**Limitations:**

Given that DeRy combines different models, what about the bias of the resulting model? Is there any way to control the biases of the combined models so that they won't be reflected on the final model? While this is beyond the scope of the current paper, a discussion on limitations wrt to biases would be interesting.

**Strengths And Weaknesses:**

Strengths:

* The paper is very well written and easy to follow.

* The problem of model reassembly is interesting with many practical use-cases.

* The proposed method is novel and the two step procedure makes sense.

* Empirically demonstrated that the proposed method obtains competitive performance on ImageNet dataset.

Weaknesses:

Overall, I liked the problem definition and proposed solution to address the interesting model reassembly task. However, I would like the authors to address my following concerns/questions to further improve the quality of the work.

* Why the proposed approach focuses on fusing heterogeneous models, can it be applied to homogenous models? If yes, how does that compare to the existing model fusion methods that work only for fusing homogenous models, e.g.,  [54, 14, 55, 59]. I would suggest the authors to include experiments for this to strengthen the paper.

* How does the size of model zoo effect the performance of the proposed method? In particular, experiments and discussion along the direction of scalability of the proposed method wrt to number of models in the model zoo would be interesting.

* Can model reassembly help in reducing the training cost of the large models? A discussion on potential applications of model reassembly besides knowledge transfer would be good for demonstrating a broader impact of the work.

* Authors mention that "We manually identify the atomic node to satisfy our line graph assumption. Each network is therefore a line graph composed of atomic nodes." This is not clear from the current descriptions. Can authors explain this?

* What is the effect of partition number on the final performance? How does it increase the complexity of the method?

* How is the proposed method comparable to Model Fusion of Heterogeneous Neural Networks via Cross-Layer Alignment? A comparison and discussion should be included in the paper to verify the advantage of the method over similar methods.

* Can the proposed approach DeRy be applied to other computer vision tasks beyond image classification?

---

> ### Author Response · Authors · 2022-08-02
> **Response to Reviewer 3qgQ (PartⅡ)**
>
>
> `>>> Q4:` **Atomic Node Definition**
>
> `>>> A4:` We thank the reviewer for the comment. In our study, not every operation in the DNN can be treated as an atomic node. Consider a `Conv->ReLU` with skip-connection; we cannot make the single convolution layer as our node because a skip-connection breaks the line graph assumption. DeRy, in this current form, can not cut off multiple parallel paths at the same time. Therefore, we need to specify the node in each network. For example, a ViT-B contains 12 transformer blocks and hence has 12 nodes, and a ResNet-18 has 8 residual blocks and hence has 8 nodes. The detailed definition of atomic nodes is listed in the source code `blocklize/block_meta.py`.
>
> `>>> Q5:` **Partition Number, Performance and Complexity**
>
> `>>> A5:` We repeat the network partition and reassembly step with `K=5` and `K=6` to see how our method performs with different partition granularities. We set the configuration to `DeRy(K, 30, 6)`. The network is trained on ImageNet for 100 epochs. All other experimental settings are kept the same as those used in the manuscript. As illustrated in the table below, we observe that the partition number `K` has little effect on the model performance. As for the computational complexity, a large `K` does increase the search time with less than 10% time growth.
>
> | K | \# Param | GFLOPs | Top1 Acc | Top5 Acc |
> | - | -------- | ------ | ----------------- | ----------------- |
> | 4 | 24.89    | 4.47   | 79.63              | 94.81              |
> | 5 | 21.14    | 5.53   | 79.68             | 94.89             |
> | 6 | 23.38    | 5.39   | 79.83             | 95.02             |
>
> `>>> Q6:` **CLAFusion Comparison**
>
> `>>> A6:` We thank the reviewer for pointing out this interesting and inspiring reference [A]. Though Cross-Layer Alignment Fusion (CLAFusion) and our paper share similar high-level motivations, the problem setup and the solution are quite different. We have cited the paper in our revision in [72] and provided a discussion.
>
> 1.  **Problem setup**: CLAFusion also aims to fuse heterogeneous neural networks, but it is restricted to the case where two networks come from the same architecture family, with the same input and output dimension but different numbers of hidden layers. DeRy, by contrast, does not impose any assumption on the network structures, where CNN, MLP, and transformer can be reassembled in a unified framework.
>
> 2.  **Solution**: CLAFusion first solves a layer assignment problem between two networks, and then transforms them to the same depth through either adding or merging layers. As such, their fusion is `task-agnostic`: for any two networks, the fusion does not depend on the final task or data. We, instead, partition the networks into building blocks and then reassemble them to maximize the target performance. Hence, our method is `target-related`: the reassembled model is searched on a specific target task.
>
> 3.  **Extensibility**: CLAFusion is originally designed for fusing two networks. It is hard to extend CLAFusion to multiple models. To fuse K networks, the solution in the CLAFusion paper needs to run the pairwise fusion for K times. DeRy is, on the other hand, more scalable, since we only need to run the algorithm once, regardless of the number of models involved.
>
> [A] Nguyen, Dang, et al. "Model Fusion of Heterogeneous Neural Networks via Cross-Layer Alignment." arXiv preprint arXiv:2110.15538 (2021).
>
>
> `>>> Q7:` **Extension to Other Vision Tasks**
>
> `>>> A7:` Yes, DeRy can be indeed applied to other vision tasks. There are several advantages of DeRy when it is applied to downstream tasks.
> 1. **First**, as DeRy directly searches for a general backbone, we may readily apply the same network to other tasks without any hassle.
> 2. **Second**, the training-free proxy of NASWOT does not depend on the ground-truth label and is therefore label-agnostic. It enables us to assemble new networks on any task and any modality of input.
> 3. **Third**, DeRy is highly computationally efficient; it only requires several hours to search for the optimal structure on a large-sized dataset.
> We indeed look forward to extending DeRy to other vision tasks in our future work.

---

> > ### Author Response · Authors · 2022-08-08
> > **Thank the Reviewer for the Constructive Comments**
> >
> > Dear Reviewer,
> >
> > We would like to thank you again for your constructive comments and kind effort in reviewing our submission. Please do let us know if our response has addressed your concerns, and we are more than happy to address any further comments.
> >
> > Thanks!

---

> > > ### Comment · Reviewer_3qgQ · 2022-08-09
> > > **Final Response**
> > >
> > > I thank the authors for their response and effort on the new experiments. While most of my concerns have been addressed by the reviewers, I personally think it would be much better if authors could fix the problems I mentioned and update the manuscript accordingly, rather than just making a promise that they will add a discussion in the main paper. The new experiments and discussion presented here should be either included in the main paper or supplementary to further strengthen the proposed work.
> > >
> > > Moreover, given that DeRy combines different models, what about the bias of the resulting model? Is there any way to control the biases of the combined models so that they won't be reflected on the final model? While this is beyond the scope of the current paper, a discussion on limitations wrt to biases would be interesting.

---

> > > > ### Author Response · Authors · 2022-08-09
> > > > **Manuscript Updates and Further Clarifications**
> > > >
> > > > `>>> Q8` **Manuscript Update**
> > > >
> > > > `>>> A8` We sincerely appreciate the reviewer for the constructive feedback. As advised, we have further updated the manuscript. Specifically, we add the experimental results on the homogeneous model zoo in `Supplementary Section 5.1` and add the ablation study for partition number $K$ in `Supplementary Section 5.2`. We also provide in-depth discussions on the potential applications in `Supplementary Section 4`, the limitation on model bias in `Supplementary Section 3`, the extension to other tasks in `Supplementary Section 6`, and the node definition in `Supplementary Section 7.3`.
> > > >
> > > > `>>> Q9` **Model Bias Elimination**
> > > >
> > > > `>>> A9`
> > > > Yes, the reviewer's comment is well taken. It is indeed possible that DeRy transfers the biased knowledge from the predecessors to the reassembled model. To address this problem, in our `Supplementary Section 3`, we discuss how to resolve this issue. Specifically, two techniques could be incorporated into the DeRy framework to mitigate the model bias.
> > > >
> > > > **First**, we can expand the model zoo size and limit the block size for each model. It ensures that no single block is dominant in the reassembled model, which largely rules out the possibility of a large bias from each individual network.
> > > >
> > > > **It is also possible** to increase the diversity among the reassembled blocks instead of blindly optimizing the target performance. A diversity regularization term could be added to Equation 8 to promote unbiased predictions.
> > > >
> > > > We will extend our study to those fields to eliminate the bias introduced by DeRy in future work.

---

> > > > > ### Comment · Reviewer_3qgQ · 2022-08-09
> > > > > **Followup Response**
> > > > >
> > > > > I thank the authors and appreciate their effort in improving the manuscript. To summarize, all of my concerns are now well addressed by the authors and hence I am increasing my initial score to Strong Accept.

---

> ### Author Response · Authors · 2022-08-02
> **Response to Reviewer 3qgQ (Part I)**
>
> We would like to thank the reviewer for their insightful feedback and interesting observations. We are encouraged that the reviewer found the problem setup novel, the experiments thorough, the proposed method well-designed, and the paper well-written and easy to follow. We thank the reviewer for the support of our work. We address the reviewer’s comments below and will include all the feedback in the revised version of the manuscript.
>
> `>>> Q1:` **Heterogeneous Models VS Homogeneous Models**
>
> `>>> A1:` We thank the reviewer for the question. Yes, DeRy can indeed be applied for homogeneous models, since, in fact, a homogeneous model zoo is a particular and simplified case for heterogeneous models. In our manuscript, we have focused on the more challenging heterogeneous models. As advised by the reviewer, here we adopt DeRy on homogeneous models and compare the results with those of [54] on CIFAR-100, AirCraft, and Cars, using the same homogeneous model zoo settings as in [54]. The approaches of [55, 59] are, however, explicitly designed for federal learning and hence do not fit our goal; the approach of [14], on the other hand, had no open-sourced implementation online.
>
> Note that we do not further pre-train DeRy on ImageNet to make sure the comparison is fair. As shown below, we indeed outperform [54] significantly with the same experimental setup.
>
> |  | # Param |CIFAR-100 | AirCraft| Cars|
> |--|--|--|-- |-- |
> |Zoo Tuning [ICML 2021]  | 23.71 |83.39 |85.51 | 89.73|
> |DeRy(4, 30, 6)|24.89 | 84.05(**+0.66**) |88.86(**+3.35**) |93.86(**+4.13**) |
>
> `>>> Q2:` **Model Zoo Size and DeRy Scalability**
>
> `>>> A2:` We truly thank the reviewer for the comment. Due to the computational constraints, we did not compare the performance of different model hub sizes. We will extend our experiment in that direction in future work.
>
> But a quick thought experiment is that, since our method partitions and reassembles the model zoo in an AutoML style, scaling up the model zoo is truly nothing beyond increasing the feasible set size. Therefore, the global optimal reassembly could ideally perform better. The only problem is that scaling up the feasible set makes the optimization harder, which requires transversing more candidates to reach the optimality. As our zero-shot proxy is very cheap in terms of computation, we believe our method generalizes well to large model zoos.
>
> `>>> Q3:` **Potential Applications and Reducing the Training Cost**
>
> `>>> A3:` Thanks for the question. DeRy indeed has a wide range of potential application scenarios. Taking the foundation model as an example, instead of training a network from scratch, we can take several pre-trained small models, partition them into building blocks, and assemble them into the large model as an efficient method for network initialization. As suggested in the paper, reassembling pre-trained models provides faster convergence and reduces the training cost.
>
> Another example is multi-task learning. Given a bunch of trained single-task models, we can bring up a method to aggregate their capacities into a reassembled model. For example, assemble a new model with a shared backbone and multiple task prediction components, each taken from a single task. As such, we reassemble a multi-task model at a very low cost using DeRy.
>
> As suggested by the reviewer, we will add a discussion in the main paper.

---

### Official Review · Reviewer_k5P1 · 2022-07-12

**Rating:** 8
**Confidence:** 4
**Soundness:** 4 excellent
**Presentation:** 4 excellent
**Contribution:** 4 excellent

**Summary:**

This paper brings up an ambitious and pioneering paradigm, termed Deep Model Reassembly (DeRy), that reuses the pretrained neural networks as building blocks for new model construction to address the transfer learning problem. A two-stage solution is proposed to jointly search for the optimal model architecture and weight for the reassembled model. DeRy first partitions the pre-trained networks jointly via a cover set optimization, and then assemble blocks to customize networks subject to hard constraints via solving an integer program backed up with a training-free proxy. Experimental results on ImageNet and 8 down-stream tasks validate that the reassembled model can achieve higher performance than any candidate models in the model zoo.

**Questions:**

- Add more literature review on NAS. Given that there is a focus on joint search for architecture and model weights with zero-cost proxy in this paper, the literature review should be extended beyond a handful of papers currently listed. What is the main difference between the proposed DeRy and the traditional NAS problem?

- Although the staged solution largely eliminates the combinational search for a search space, however, the authors pose strong heuristics in the model partition stage. The blocks with high ``functional similarity'' are grouped and the partition should maximize the overall group-ability. Why the ``functional similarity'' introduced in this paper gives rise to swap-ability?

- Typos:
1. Line 170: a path graphs -> a path graph
2. Line 179 and Eq (3): What does N_g stand for? Is it the number of equivalence sets? Should it be the same as K?
3. Line 207: process -> possess
4. Line 229: architecture -> architectures



**Limitations:**

Yes, limitations have been discussed. I do not find the potential negative societal impact of this work.

**Strengths And Weaknesses:**

[Strengths]
- Overall, this is a well-written paper with an interesting and extensible idea at its core: considering the knowledge transfer as finding optimal layer-wise stacking of pre-trained blocks (In Definition 1). Comparing to the typical NAS problem with a fixed search space like operation or topology, the DeRy involves a dynamic search space (The partition of the pre-trained networks is unknown beforehand), which is much more challenging.

- New Findings: This study strives to point out that, arbitrary trained networks are largely reassembleable, even though the models may have diverse architecture and source tasks. The grafted pre-trained models potentially could provide satisfying performance.

- Overall, I am convinced with the intuition of the method: first dissecting the networks into swappable equivalence blocks, then constructing new models with the best performance. The dedicated solution draws inspiration from the conventional discrete optimization community, thus with a good convergence guarantee and low search complexity. I personally favor the current solution.

- Evaluation is sufficient to support the argument of this paper. Section 4.1 validates the pipeline designs and lists several intriguing findings for partitioned blocks

[Weaknesses]
- If my understanding is correct, the current methodology may result in an over-complicated pipeline for the model reassembly, which may not be scalable to more sophisticated tasks (detection, segmentation and generation) or model zoo setting. I am not expecting the authors to resolve this issue in this paper, but it would be great to have a discussion over this matter in future work part.

- The current solution still needs a long training time, like 100~300 epochs for ImageNet. This is OK since this is the first attempt along the line, but again it would be great if the authors can provide a discussion on this issue.

- Despite the authors already including quite a significant number of experiments, it would be better if more pretraining tasks and datasets be included in the evaluations.

---

> ### Author Response · Authors · 2022-08-02
> **Response to Reviewer k5P1**
>
> We thank the reviewer for the constructive comments and would like to address them as follows.
>
> `>>> Q1` **Over-complicated Pipeline and Scalability**
>
> `>>> A1` Thanks for the question. In fact, DeRy introduces an acceptable complexity and a low computational cost; also, it can be easily extended to more complicated vision tasks other than classification. The reason lies in the following aspects. First, DeRy searches for a general vision backbone, so it's easy to apply the same network to a handful of tasks. Second, NASWOT, which is adopted in DeRy, is a task-agnostic proxy, and it allows us to build customized networks on arbitrary tasks or input modalities. Lastly, DeRy has a low computational overhead: the search time is no longer than 0.5 GPU day. Hopefully, we will be able to extend DeRy to other applications in the future.
>
> `>>> Q2` **Training Time**
>
> `>>> A2` Thank the reviewer for the question. For a fair comparison, we use a training schedule of 100~300 epochs in accordance with the setting in other works. In fact, one advantage of DeRy is that it converges faster than randomly initialized networks. As demonstrated in Table 9 and Figure 10, our model is superior to baselines with faster convergence, where the loss function decreases rapidly. In section 4.1 `Similarity, Position and Reassembly-ability` section, we also experiment with training the reassembled model for 20 epochs. The best-reassembled network (ResNet50 with stage 3 replaced by ResNet101) also archives a competitive top-1 accuracy of around 78% on ImageNet. These results indeed demonstrate that DeRy enjoys faster convergence.
>
> We sincerely hope that the reassembled model may perform well with very little training time (less than 10 epochs or 1 epoch) or even zero-shot reassembly since all network blocks have previously been trained. However, we are still far from achieving this ambitious goal. We look forward to handling this issue in future work.
>
> `>>> Q3`  **Diverse Pre-training Tasks and Datasets**
>
> `>>> A3`  Thanks for the comment. In the long run, we would like to reassemble models in a larger, more diverse model zoo. In spite of this, we are not able to test our method on a larger scale due to computational limitations. Secondly, most of the model zoos available online are based on ImageNet, which is quite homogeneous and difficult to use. To eliminate data bias, we will explore collecting a more diverse model zoo in the future.
>
> `>>> Q4`  **Comparison between DeRy and NAS**
>
> `>>> A4` There is a significant difference between DeRy and NAS
>
> 1.  **Partition Step**. Initially, DeRy subdivides a group of neural networks into blocks and then reassembles them into a customized network. The NAS, on the other hand, assumes that the search space has been predefined, so partitioning is not needed.
>
> 2.  **Distinct Objective**. While DeRy searches for the architecture and weights jointly, NAS concerns only the network architecture.
>
> We truly thank the reviewer for the suggestion. We will consider adding a section on NAS and its connection with our study.
>
> `>>> Q5` **Functional Similarity and Swap-ability**
>
> `>>> A5` Thank the reviewer for the comment. Here's a quick thought experiment. Consider the *Linear Regression* as our similarity measurement. A large functional similarity of the two blocks indicates that both input $X, X’$, and output features $B(X), B’(X’)$ are highly similar. Consequently, a simple linear layer $F$ is able to transform one input into another input $X’\approx F(X)$. The same applies to the outputs $B(X) \approx F’(B’(X’) )$ with another linear transform $F'$. Therefore, we can replace $B \approx F\circ B’ \circ F’$, where $\circ$ stands for network stacking. According to the kernel functions employed in the similarity computations, different stitching layer structures could be specified to satisfy the swap-ability requirement.  Hope our answer addresses the reviewer's question.
>
> `>>> Q6` **Writing Problems**
>
> `>>> A6` We truly appreciate the reviewer for the proofreading. We have fixed these typos in the revision.

---

> > ### Author Response · Authors · 2022-08-08
> > **Thank the Reviewer for the Constructive Comments**
> >
> > Dear Reviewer,
> >
> > We would like to thank you again for your constructive comments and kind effort in reviewing our submission. Please do let us know if our response has addressed your concerns, and we are more than happy to address any further comments.
> >
> > Thanks!

---

### Meta-Review · Area_Chair_Udkv · 2022-09-03

**Recommendation:** Accept
**Confidence:** Certain

**Metareview:**


 This paper proposes an interesting new way to think about how to use a model zoo of pre-trained models: extract modular building blocks that are swappable from the networks and then stitch them together. To do the former, a cover set optimization methods is proposed, and the blocks can then be combined in a way that respects various resource and performance constraints. The idea is both interesting and ambitious, and has the potential to open up various avenues of research if done well.

The paper lives up to the task: It is well-written (k5P1, 3qgQ, q2yK), conducts experiments to validate if the such stitched networks can do well, and proposes an intuitive principled method to extract the blocks (3qgQ, k5P1). The reviewers did express some concerns about scalability/generalizability to other tasks (k5P1, 3qgQ), larger zoos (k5P1 ,3qgQ), other architectures (all reviewers), and computation (q2yK) as well as several other potential issues such as limited performance improvements. The authors provided strong rebuttals to these, including some new experiments. At the end of the process, the reviewers were all satisfied with most of the concerns, and the overall consensus on the paper seems to be with high scores.

  Given the potentially high-impact, novel perspective as well as the solid execution, I highly recommend this paper for acceptance.

**Award:**

Yes

---

### Decision · Program_Chairs · 2022-09-14

Accept